# Towards an Automatic Early Warning System of Flood Hazards based on Precipitation Forecast: The case of the Miño River (NW Spain)

J. González-Cao, O. García-Feal, D. Fernández-Nóvoa, J.M. Domínguez-Alonso, M. Gómez-Gesteira

Environmental Physics Laboratory (EPhysLab), CIM-UVIGO
Universidade de Vigo
Ourense, Spain

*Correspondence to*: D. Fernández-Nóvoa (diefernandez@uvigo.es)

**Abstract:** An Early Warning System for flood prediction based on precipitation forecast is presented. The system uses rainfall forecast provided by MeteoGalicia in combination with a hydrologic (HEC-HMS) and a hydraulic (Iber+) model. The upper reach of the Miño River and the city of Lugo (NW Spain) are used as a study area. Starting from rainfall forecast, HEC-HMS calculates the streamflow and Iber+ is automatically executed for some previously defined risk areas when a certain threshold is exceeded. The analysis based on historical extreme events shows that the system can provide accurate results in less than one hour for a forecast horizon of 3 days and report an alert situation to decision-makers.

## 1. Introduction

According to Noji (2000), floods are one of the most dangerous natural hazards in the world. Jonkman (2005) estimated that more than 100,000 deaths in the last century were caused by floods. From 1940 to 2018 the number of deaths related with flood events (8,138) is only surpassed by the lightning fatalities (9,386) in the U.S. (https://www.weather.gov/hazstat/). Furthermore, the effect of the Climate Change will increase the number of flood events and their negative impact to people and properties (Dankers and Feyen, 2008; Alfieri et al., 2017). Therefore, the ability to predict these extreme events and prevent their consequences is a challenge for the scientific community worldwide.

In this context early warning systems (EWS) play a key role. UNISDR (2009) defines early warning systems as "the set of capacities needed to generate and disseminate timely and meaningful warning information to enable individuals, communities and organizations threatened by a hazard to prepare and to act appropriately and in sufficient time to reduce the possibility of harm or loss". A complete EWS is divided into four steps: (1) risk knowledge, (2) monitoring, forecasting and warning, (3) communication of an early warning system and (4) response capability (UN, 2006). The first two steps are related to the field of physical sciences while the two last steps are associated to social science aspects. There are several works related to the impact of the early warning system in the prevention of floods. Baudoin et al. (2014) and UNISDR (2015) show some interesting examples on how early warning systems can save lives and reduce the damage to the people. Borga et al. (2011) developed an early warning system methodology for flash floods in Europe through the HYDRATE project. The authors enhanced the capability of flash flood forecasting in ungauged basins by exploiting the extended availability of flash flood data and the improved process understanding. Alfieri et al. (2012) analysed several early warning systems applied to detect surface water flooding, flash floods, debris flows, land-slides induced by extreme rainfall events, river and coastal floods. The authors proposed several tasks to palliate the main drawbacks of some of these systems. Also, Hossain et al. (2014) developed a system to measure the water depth of the river at the "Valley of Death" and Cools et al. (2012) developed an early warning system to detect flash floods in the Sinai Peninsula, both based on a satellite-based forecast system. In Europe a very interesting example of an early warning system is the EWS applied to the region of Flanders (Schelfaut et al., 2012 and CIW, 2013). In this work, the different steps are analysed under the FREEMAN project (Flood REsilience Enhancement and MANagement). The European Flood Awareness System (EFAS) is also another example of an EWS developed to the sponsorship of the European Commission. This system provides daily streamflow forecast for Europe starting from up to 10-days weather forecast (medium-term forecast). More details of this model can be shown in Thielen et al. (2009), Pappenberger et al. (2011), Cloke et al. (2013) and Alfieri et al. (2014). Using this model Dottori et al. (2017) develop a methodology to adapt EFAS to real time forecasting. Demerit et al. (2013) analyse the problems derived from the use of the early warning system to medium and long-term flood forecast, mainly the dissemination of the information to people potentially affected by these events. They reveal that flood forecasters usually wait the confirmation from local institutions

(Hydrologic Confederations…) instead of acting following the information provided by the early warning systems. These local systems are focused in short-term forecast (0 to 48 h) that are more suitable to evacuation than fore damage mitigation. Some examples of these short-term local systems focused on river floods are: the River Forecast Centers (https://water.weather.gov/ahps/rfc/rfc.php) in the United States of America or "Sistema de Ayuda a la Decisión" (http://www.chebro.es/contenido.visualizar.do?idContenido=12789&idMenu=2902) developed by the Hydrographic Confederation of the Ebro river (Spain). In Europe the meteoalarm (http://www.meteoalarm.eu/?lang=en_UK) provides advice on exceptional weather events including floods with a temporal window of 48 h. There are mainly two kind of floods derived from precipitation events: flash-floods and river-floods. On the one hand, flash-floods are characterised by a delay time, from the peak precipitation time to the peak of flood, from 3 to 6 hours. These floods are usually registered in dry climate and rocky terrain areas due to the lack of vegetation to infiltrate the precipitation into the ground. These kind of floods have associated a very high level of risk due their velocity of propagation. On the other hand, river-floods are generally registered in larger rivers in areas with a wet climate and the delay time is greater than 6 hours. The consequences associated to the latter ones can be also dramatic to the people and their properties. This make necessary to develop an EWS to improve the security of the areas exposed to these events. The area of study analysed in this work is mainly affected by river-floods.

In this paper, a flood early warning system based on precipitation forecast is presented. The system, which is being developed in collaboration with the Hydrographic Confederation of Miño-Sil River, consists of three steps: i) precipitation forecast; ii) use of a hydrologic model to predict extreme flows; iii) use of a hydraulic model that is applied at certain areas only under extreme flows. Starting from 1-day, 2-day and 3-day precipitation forecast windows provided by the Regional Meteorological Office (MeteoGalicia), the outflows associated to the catchment of the Miño River (NW Spain) were obtained using the HEC-HMS model (U.S. Army Corps of Engineers, 2018). This model was calibrated for the area of study by means of series of historical flood events detected over the last decade. The numerical model Iber (Bladé et al., 2014) was used to obtain water depth and velocity under extreme flow conditions for some risk areas where previous events have caused damages or material loses. Both models (i.e., HEC-HMS and Iber) are freely available software so the system can be applied at any location without

costs derived from the licences of commercial codes. The main contribution of the EWS
presented in this work respect to the systems shown in the bibliography is that all the
components are freely available and easily adaptable to different areas of the world.
The paper, which aims to describe the steps followed to develop the EWS, is organised
as follows. First, a description of the area of study (the upper reach of Miño River and the
city of Lugo, NW Spain) is shown. Then the methodology to obtain the weather forecast,
the computation of the run-off and the hydraulic processes are briefly presented. Also the
communication among all the models (Precipitation Forecast - Run-Off - Hydraulic
processes) is explained. Next, the results of the precipitation and outflow forecast of a
series of historical flood events are presented along with a statistic analysis of their
accuracy. Finally, the numerical water depth obtained for a particular flood event at the
city of Lugo is shown and compared with field data measured during the event.
**2. Study area**
The area of study is located in north-western Spain (Figure 1). It corresponds to the upper
reach of the Miño River. This sub-catchment area is about 2200 $km^2$ and the elevation
ranges from 360 to 980 m.a.s.l. The average annual precipitation ranges from 144 to 1300
mm $year^{-1}$. Miño River presents an annual hydrologic cycle characterised by a pluvial
regime, presenting maximum river discharges during winter months descending then to
reach its minimum values during summer (Fernández-Nóvoa et al., 2017). Specifically,
considering the period under study at Lugo station, Miño River reaches maximum flows
of 114 and 128 $m^3s^{-1}$ in January and February and minimum ones of 7 and 8 $m^3s^{-1}$ in
August and September, respectively.
Figure 1 (upper-left panel) shows the catchment of the upper reach of the Miño River,
which is divided into three main sub-basins according to their topographic characteristics.
Seven rain gauges operated by MeteoGalicia are located in the entire sub-catchment.
Table 1 shows the location and the elevation of each of the rain gauges located in the
upper reach of the Miño River. The outlet of this catchment is located in the city of Lugo
(Figure 1, lower panel). This area is usually flooded during the events of extreme
precipitations in the upper reach of the Miño River. The absence of dams in the catchment
to regulate the flow also affects the high frequency of these events.
**3. Methodology**

In this work, an automatic EWS is proposed. This system is composed of several elements as shown in Figure 2. All these components are orchestrated by a Python script that is the responsible of gather and transform the data properly in order to feed the models used in the system. First of all, the rainfall forecast performed with the Weather Research and Forecasting model is provided by the weather agency (MeteoGalicia). Details are provided in next section. Forecasted data are automatically downloaded and the rainfall relative to each sub-basin is extracted to fed the hydrological model HEC-HMS. When the catchment outflow obtained with HEC-HMS surpasses the 90th percentile of historical data, it is considered as a possible extreme event and the following steps will be applied. In that case, this outflow will be used as inlet condition for the hydraulic simulation using the model Iber to provide flood maps with water depths and velocities at certain risk areas (the city of Lugo in this particular case). Data provided by Iber are processed for hazard evaluation. At this stage the system checks if there is a risk condition in the areas accessible by pedestrians. These areas are user defined and can be changed depending on seasonal events. In order to emit a warning alert, the criteria of Cox et al. (2010) are used to define safety limits for children since they are the most vulnerable population group. Following this criterion, a warning will be emitted if there is a zone where any of the following thresholds are surpassed: the water depth ($h$) is higher than 0.5 m, the magnitude of water velocity ($v$) is higher than 0.2 ms$^{-1}$ or the product ($h·v$) excess 0.4 m$^2$s$^{-1}$. This warning is sent in form of report to a decision maker, so an expert can validate the resulting data and discard false positives.

The details of the components of the EWS, the data sources, and the calibration processes are described in the following sections.

**3.1 Precipitation data**

**3.1.1 Forecasted precipitation data**

Forecasted precipitation data were obtained from the Regional Meterological Office (MeteoGalicia, http://www.meteogalicia.gal/). MeteoGalicia publishes weather forecast results based on the Weather Research and Forecasting (WRF) Model (Skamarock et al., 2005) (https://www.mmm.ucar.edu/weather-research-and-forecasting-model). The WRF model is a numerical weather prediction system at regional mesoscale designed mainly for forecasting applications. WRF is run operationally since 2008 providing daily data until the end of 2012 (00 UTC) and twice a day (00 UTC and 12 UTC) from then on, with

a 72 hour forecast window, a temporal resolution of 1 hour and maximum spatial
resolution of 4 km (Sousa et al., 2013). Data provided by MeteoGalicia are freely
available at its THREDDS (Thematic Realtime Environmental Distributed Data Service)
server, also maintaining an historical archive of past forecast since 2008. The model
outputs provide several variables related to weather. In the case of this study, precipitation
information was automatically obtained for the areas under interest at the 00 UTC of each
day during the period 2008-2018.
### 3.1.2 Measured precipitation data
Real precipitation data at hourly scale were obtained from the rain gauges managed by
MeteoGalicia, which is responsible of their maintenance and data quality control. Data
from these rain gauges was used to assess the performance of the MeteoGalicia Weather
Forecast to predict extreme rain events. The mentioned rain gauges are pictured in Figure
1 and their location and elevation is detailed in Table 1.
### 3.2 River discharge data
Daily discharge data of the Miño river were provided by the corresponding river Basin
Authority (Confederación Hidrográfica del Miño-Sil, https://www.chminosil.es). In this
case of study, Miño flow data at Lugo station covering the period 2008-2018 were
selected. River data were used to calibrate and validate the hydrologic model system used
during the development of this study.
### 3.3 HEC-HMS & Iber+
Here the hydrological and hydraulic models used in the study will be briefly described
along with the methods to analyse their accuracy.
The semi-distributed model HEC-HMS (Feldman, 2000 and U.S. Army Corps of
Engineers, 2018) was used to analyse the rain-runoff processes and the numerical model
Iber (Bladé et al., 2014) was used to compute the hydraulic processes.
The HEC-HMS is a model developed by the US Army Corps of Engineers that is applied
to simulate continuous hydrological processes. The HEC-HMS model can be used to
analyse various hydrological aspects, such as flooding events, reservoir capacity,
stormwater warnings, and stream restoration (U.S. Army Corps of Engineers 2008).
HEC-HMS is divided into four components: (i) an analytical model: calculation of direct
runoff and channel routing; (ii) a basin model: representation of hydrological elements in
a watershed; (iii) a system to manage input data and store data; (iv) a post-processing tool
to report and illustrate simulation results. Two main processes were taken into account in
the methodology developed in this case of study: loss (infiltration) and transform
methods. In the first case, the Soil Conservation Service (SCS) curve number was
selected. This method implements the curve number methodology for incremental losses,
since it was designed to calculate the infiltration during periods of heavy rainfall, and
therefore is well suited to this type of studies. Respect to the transform process, based on
the way of convert the excess precipitation as runoff, the SCS unit hydrograph method
was also selected for the reasons mentioned above. More information about the loss and
transform methods used in this work are detailed in NRCS (2007). By last, the
Muskingum-Cunge Routing method was selected for runoff propagation because it
provides a good approach in basins with similar slopes. This method takes into account
the conservation of mass as well as the diffusion representation of the conservation of
momentum (U.S. Army Corps of Engineers 2008). Other parameters like the baseflow
were not considered because suppose less than 3% of the peak flow for this kind of events
and can be neglected.
Taylor diagrams (Taylor, 2001) were used to compute the accuracy of the results obtained
with HEC-HMS by means of the normalised standard deviation (Eq. 1), normalised
centred root-mean square difference (Eq. 2) and correlation (Eq. 3).

$$\sigma_{n,A} = \frac{\sqrt{\frac{\sum_{i=1}^{N}(A_i - \bar{A})^2}{N}}}{\sigma_B} \tag{1}$$


$$E_{n,A} = \frac{\sqrt{\frac{\sum_{i=1}^{N}[(A_i - \bar{A}) - (B_i - \bar{B})]^2}{N}}}{\sigma_B} \tag{2}$$


$$R_A = \frac{\sum_{i=1}^{N}[(A_i - \bar{A})(B_i - \bar{B})]}{N\,\sigma_A\,\sigma_B} \tag{3}$$


where $A$ is a numerical variable and $B$ a reference variable. The subscript $n$ refers to the
normalised parameter, subscript $i$ refers to the different samples, $N$ is the number of
samples, barred variables refer to mean values and $\sigma$ is the standard deviation.
The hydraulic simulations were carried out using the numerical model Iber (Bladé et al.
2014). Iber is a numerical code that solves the 2D (Two-Dimensional) Shallow Water
Equations by means of finite volume schemes (FVS). The software package is formed by
three elements: pre-processing tool, numerical model and post-processing tool. The first
and the last modules are based in the software GID (GID, 2018). It provides a user
friendly graphical interface (GUI) to create the case and edit the parameters that define
the problem to solve. It also provides tools to analyse the results of the numerical
simulations. The pre-processing and post-processing tools were used only during the
modelling and testing of the study area. However, the automatic EWS runs the model in
batch mode without user interaction. Iber was recently improved in terms of efficiency
becoming Iber+ (García-Feal et al. 2018). This new parallel implementation of the Iber
model takes advantage of GPU (Graphics Processing Unit) computing using the Nvidia
CUDA (NVIDIA CO., 2019) platform. Using this technology, the new implementation is
able to run up to 100 times faster. This fact makes Iber+ especially suitable for the
implementation of an EWS where the response times can be crucial to issue an early alert.
The accuracy of the water depth results computed with Iber+ at five control points was
assessed by means of the *bias* and the *RMSE* (Root Mean Square Error) for the extreme
event recorded on January 2013

$$RMSE = \sqrt{\frac{\sum_{i=1}^{N}(A_i - B_i)^2}{N}} \qquad (4)$$

$$bias = \frac{\sum_{i=1}^{N}(A_i - B_i)}{N} \qquad (5)$$

where $A$ is numerical value, $B$ the measured value and $N$ the number of control points.

## 4. Results and discussion

### 4.1 Accuracy of MeteoGalicia Precipitation Forecast

The capability of MeteoGalicia Weather Forecast system to predict rain events was
evaluated by means of the comparison with real precipitation data provided by the rain
gauges in the area of study. For that purpose, the predicted (numerical) precipitation was
obtained at the closest grid points to the location of the rain gauges. The correlation
between predicted and measured precipitation was calculated for each rain gauge during
the available period (2008-2018). For this calculation, Spearman rank correlation was
used due to its robustness to deviations from linearity, as well as its strength to the
influence of outliers. This procedure was carried out for 3 forecast windows (1-24 h, 25-
48 h and 49-72 h; 1-day, 2-day and 3-day forecast from now on) to determine the accuracy
of the forecast at different temporal scales. The comparison is carried for an aggregation
time of 24 h, which matches the recording frequency of rain data provided by
MeteoGalicia and is compatible with the kind of flood events (mainly river floods) of the
area.
The values of the correlation and the normalised standard deviation for each rain gauge
are shown in tables 2 and 3. Table 2 shows the analysis for the complete series and table
3 shows the results considering only rainy events (precipitation above the 75 percentile).
In general, considering the complete series, precipitation prediction offers a good
representation of the registered values and the variability of precipitation. In fact,
correlations above 0.8 were obtained for the first two windows (1-day and 2-day forecast),
although with a higher correlation for the first one. The correlation is slightly lower for
the 3-day forecast, although it is still close to 0.8. When only rainy events are considered
mean correlation values are slightly lower than considering the complete series, although
showing a good representation of the registered data. It is specially remarkable the high
correlation showed under 1 day forecast window with a mean value above 0.7 (Table 3).
Respect to the normalised standard deviation, most of cases in both series are similar to
1, which shows a good agreement between forecast and real precipitation. Therefore, it
can be concluded that the precipitation forecast provided by MeteoGalicia offers results
very close to the real rain events for the entire time series of precipitation data (2008-
2018). This shows the accuracy of MeteoGalicia models to forecast precipitation events
up to three days in advance.
**4.2 Calibration and validation of hydrological processes using HEC-HMS**
A set of 15 extreme flood events registered during the period 2008-2018 were used to
calibrate and validate the rain-runoff model HEC-HMS (Table 4) by comparing the
outflows measured at the gauge station located at Lugo with the flows obtained with
HEC-HMS using the 1-day forecast of precipitation. Forecasted rain data were considered
because they are used to feed the model in its forecast version. In situ data would be only
valid for hindcast purposes. Calibration was carried out using the specific calibration tools
implemented in HEC-HMS (Feldman, 2000) in order to choose two independent
parameters, the curve number ($CN$) and lag time ($L_g$), for each sub-basin. The values of
$CN$ and $L_g$ were computed using particle swarm algorithm (Kennedy and Eberhart, 1995,
Pedersen, 2010 and Mezura-Montes and Coello, 2011) to minimise the error between the
measured streamflow and the numerical one. No empirical formulas were used for $CN$
and $L_g$ due the uncertainty associated to their definition (Fang et al., 2008; Upegui and
Gutierrez, 2011; Grimaldi et al., 2012). Eleven flood events were used for calibration
purposes and the rest of cases were used to validate the model. Table 5 shows the values
of the $CN$ and $L_g$ for each sub-basin obtained for each event used in the calibration step.
The mean values of $CN$ and $L_g$ of each sub-basin were used to validate the model in four
flood events (01/2013, 01/2014, 02/2016 and 03/2018) by means of a Taylor diagram
(Figure 3).
The values of normalised standard deviation ($\sigma_n$) range from to 0.8 to 1.2, the values of
the root mean squared difference (RMSD) range from 0.3 to 0.6 and the correlation of the
numerical results range from to 0.85 to 0.95. The values of $\sigma_n$ means that the variability
of the numerical results are quite similar to the variability of the reference time series
(difference less than the 20%) and the values of $E_n$ can be considered as good values
according to Moriasi et al. (2007). These values of $\sigma_n$, $E_n$ and correlation show that the
mean values of $CN$ and $L_g$ obtained in the calibration step characterise the behaviour of
the basin with a high accuracy.
Figure 4 compares the numerical and measured streamflow for the event that happened
in January 2013 using the three forecast windows. The left panel shows that time series
of the flows predicted by the model are similar to those measured at the gauge station.
The right panel is the Taylor diagram corresponding to the three forecast windows. The
standard deviation is observed to range from 0.8 to 1.2 for the three forecasts. RMSD
values for 1-day and 2-day forecasts are around 0.3, being around 0.6 for the 3-day
forecast. Finally, the correlation coefficient for 1-day and 2-day forecasts are close to
0.95, being around 0.85 for the 3-day forecast.
**4.3 Case of study**
Once the predicted water flow showed to reproduce the real events with a high accuracy
($E_n$ ~ 0.8, $\sigma_n$ ~ 0.3 and $R$ ~ 0.95), the water depth and velocity during the flood event that
affected Lugo on 20[th] January, 2013 were computed using the numerical code Iber+
(Garcia-Feal et al., 2018). Figure 5 shows the numerical domain at Lugo, where seven
land uses were defined to model the characteristics of the terrain. The Manning's
coefficient associated to each land use are shown in Table 6. Figure 5 also shows the
location of the inlet and outlet boundary conditions. The initial water depth was obtained
from data provided by the gauge station located at Lugo. The inlet condition was defined
by means of the input hydrograph (Critical/Subcritical) and the outlet condition was
defined using a supercritical/critical outflow. Turbulence was not taken into account as
suggested by (SNCZI, 2011) and according with similar works (Erpicum et al., 2010; Liu
et al. 2013; Segura-Beltrán et al., 2016).
The topography of the area of study was obtained from raster files freely downloaded
from the Instituto Geográfico Nacional website (https://www.ign.es/web/ign/portal). The
computational domain was discretised using a mesh with near 200,000 unstructured
triangular elements, with an average area of 2 m$^2$.
Five control points were defined at the area of study (see Figure 6) to analyse the accuracy
of the numerical results. Points from 1 to 4 are located in places next to the riverbank
usually frequented by pedestrians while the last one is located in the riverbed. Therefore,
the first four points are of special interest to issue an alert.
Figure 7 shows the values of the water depth obtained in the numerical simulations along
with the water depth obtained at the control points during the flood event. These field
values were obtained from photographs provided by volunteers and local media and taken
within the interval 12:00 – 16:00 on January 20$^{th}$. The numerical water depth is expressed
in terms of a mean value and a range that corresponds to 3 times the standard deviation
of the values within that interval. Visually, the numerical results are quite similar to the
field data when considering the 1-day forecast, especially if one considers that the
accumulation of the small inaccuracies of the three models involved can give rise to
biases. The values are slightly less accurate when considering the 2-day forecast and
worse for the 3-day forecast due to lower accuracy in rainfall forecast. Finally, it must be
mentioned that the depicted values do not correspond to the peak flow that took place on
21$^{th}$ January, 2013 (at approx. 4:00).
Apart from the visual comparison, the accuracy of the model to calculate water elevation
was analysed in terms of two estimators (*RMSE* and *bias*) computed using the three
forecast windows. The minimum values of *RMSE* and *bias* are obtained with the 24h
forecast window (21 cm and 0 cm, respectively). The RMSE is satisfactory when
compared with the mean upward displacement of water during the event, which is about
2.5 m. In addition, the bias is null, showing that the model (in average) neither
overestimate nor underestimate real water elevation.  The accuracy decreases with the
forecast window, although results are still good for a 2-day forecast (RMSE= 28 cm and

bias =4 cm). Finally, the accuracy is acceptable for a 3-day forecast (RMSE= 41 cm and bias =-35 cm), although with limitations in terms of bias, since the model clearly tends to underestimate field measurements. In summary, the agreement between measured and computed values indicates that the system can be used to issue alert up to 3 days in advance.

Figure 8 shows the maximum water depth and maximum velocity obtained for 1-day forecast. Hazard maps (Figure 9) can be computed from these data according to the criterion of Cox et al. (2010). Several recreation areas near the riverbanks show to have surpassed the aforementioned hazard threshold. Therefore, decision-makers can use the map to restrict activities in these areas, in order to mitigate the consequences of floods.

## 5. Conclusions

In this paper an Early Warning System for flood prediction using precipitation forecast was presented. This system starts automatically using rain forecast data retrieved from Regional Meteorological Office (MeteoGalicia) and concatenates two freely available software packages (HEC-HMS and Iber+). The upper reach of the Miño River (NW Spain) and, in particular, the city of Lugo were used as a benchmark.

A Python script was developed to deal with all the components involved in the system without user interaction. First, the precipitation forecast provided by MeteoGalicia is automatically obtained for the area of study. Second, rain forecast is provided to HEC-HMS as an input to compute the streamflow in the catchment area. When the streamflow obtained with HEC-HMS surpasses the 90th of the historical percentile at some previously selected risk area (the city of Lugo in this particular case), the possibility of an extreme event is detected and that streamflow is automatically defined as an inlet condition for Iber+. Finally, data obtained from Iber+ are processed for risk assessment and, if applicable, decision makers are reported.

The accuracy of the different models was assessed to analyse the capability of the system to provide reliable results. First, the accuracy of the precipitation forecast provided by MeteoGalicia was analysed for the period 2008-2018 showing that the 1-day forecast is slightly more accurate than the 2-day forecast, being the 3-day slightly worse, although the three forecast windows showed a reasonable agreement with field data. As a second step, the accuracy of HEC-HMS to reproduce extreme flows was assessed by means fifteen flood events recorded over for the period 2008-2018. Taylor diagrams were used

to compute the accuracy of the numerical streamflow compared with field data obtained
at the control station located near Lugo. Once again, results were satisfactory for the three
forecast windows, especially for the 1-day and 2-day forecast. Finally, a historical flood
event recorded in January, 2013 was used to assess the accuracy of Iber+ to reproduce
real water elevation at 5 control points located at the riverbank and riverbed. Both the
*RMSE* and the *bias* between the measured and computed elevations were satisfactory,
especially for the 1-day forecast.
The system needs less than 1 hour to run the models for a 3-day forecast horizon. While
data can be downloaded in a few seconds and the hydrologic model can be run in less
than a minute, no matter the extent of the area, the real bottleneck in the system is the
hydraulic model. Fortunately, the execution time does not necessarily increase with the
number of risk areas since different areas can be run concurrently when the available
hardware resources allow it. Taking into account that meteorological data are available
every day at 5:00 a.m. the system can provide an alert report to decision makers before
6:00 a.m. Additional improvements can be applied without additional cost in term of
runtime. For example, an ensemble approach can be applied when rain forecasts from
different sources are used as an input condition for HEC-HMS, in such a way that Iber+
is only executed when at least one of the hydrological realizations indicates a possible
extreme event.
Additional research is still needed to cover the entire Miño river basin, where other
problems may arise from the presence of dams. The system, when fully developed, can
even help to manage dams intelligently, maximizing energy production and dampening
floods at the same time.
The Early Warning System can be easily adapted for any area of the world since the
required input data can be obtained freely from public institutions and the models to
compute the hydrological and the hydraulic processes (HEC-HMS and Iber+,
respectively) are both freely available. Therefore, the EWS is especially interesting for
developing countries where the acquisition of commercial software is not sustainable.



*Code and data availability*. Freely available data and software (HEC-HMS and Iber+)
were used for this work. The detailed processing flowchart is shown in Fig. 2 (Section 3
– Methodology).

*Author contributions:* JGC, OGF and DFN designed the research, conducted the analysis
and wrote the paper, ; JMDA and MGG supervised the research and revised the paper.

*Competing interests.* The authors declare that they have no conflict of interest.


**Acknowledgements**
This work was partially supported by Water JPI-WaterWorks Programme under project
Improving Drought and Flood Early Warning, Forecasting and Mitigation
(IMDROFLOOD, code: PCIN-2015-243), and by Xunta de Galicia under Project
ED431C 2017/64-GRC "Programa de Consolidación e Estructuración de Unidades de
Investigación Competitivas (Grupos de Referencia Competitiva). We especially thank
Carlos Ruiz del Portal Florido, Head of the Hydrological Planning Office, Hydrographic
Confederation of Miño-Sil River for helpful discussions and for providing access to real
data within the context of INTERREG-POCTEP Programme project RISC_ML (Code:
0034_RISC_ML_6_E).

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

**Figure Captions**

**Figure 1.** Area of study. In the upper right panel, the location of the entire catchment of the shared Portuguese-Spanish river (shaded area) in the Iberian Peninsula and the riverbed of the Miño river (blue line) are shown. The rain gauges (rg1, …,rg7) located in the catchment and the sub-basins ($Sb_1$, $Sb_2$ and $Sb_3$) of the domain (upper left panel), as well as the area of study in Lugo (lower panel) are also shown. (PNOA courtesy of © Instituto Geográfico Nacional).

**Figure 2.** Flowchart of the proposed EWS.

**Figure 3.** Time series of the registered streamflow (dashed line) and numerical streamflow (orange line) of the validation events: a) 01/2013, b) 01/2014, c) 02/2016 and d) 03/2018). Taylor diagram of the validation cases are also shown.

**Figure 4.** Time series of the outflow at the control point obtained in the gauge station (dashed line) and calculated using the three forecast windows (left panel) and Taylor diagram for the same cases (right panel).

**Figure 5.** Numerical domain at Lugo. The land uses and the location of the boundary conditions (red lines) are also shown. (PNOA courtesy of © Instituto Geográfico Nacional).

**Figure 6.** Location of the five control points at the area of study in Lugo. (PNOA courtesy of © Instituto Geográfico Nacional).

**Figure 7.** Comparison between water depth ($h$ in meters) between the numerical model (.) and the field data (x) for the three forecast windows 1-day (left), 2-day (middle) and 3-day (right). The range of the numerical values correspond to 3 times the standard deviation of the elevations obtained from 12:00 to 16:00 on January 20$^{th}$, 2013.

**Figure 8.** Maximum water depth (upper panel) and maximum velocity (lower panel) obtained with Iber+ for the 1-day precipitation forecast. (PNOA courtesy of © Instituto Geográfico Nacional).

**Figure 9.** Areas where hazard criterion is surpassed. (PNOA courtesy of © Instituto Geográfico Nacional).

**Table 1.** Location and elevation of the rain gauges located in the area of study (The system of reference for latitude and longitude is the EPSG: 4326).

**Table 2.** Values of the correlation (Spearman's r) and normalised standard deviation ($\sigma_n$) of the precipitation forecast using the measured data as reference at each rain gauge, considering the complete time series of precipitation. The averaged values for each precipitation forecast are also shown.
**Table 3.** Values of the correlation (Spearman's r) and normalised standard deviation ($\sigma_n$) of the precipitation forecast using the measured data as reference at each rain gauge, considering only rainy events (above the 75[th] percentile). The averaged values for each precipitation forecast are also shown.
**Table 4**. Main characteristics of the analysed flood events
**Table 5.** Curve number (*CN*) and lag time ($L_g$) values for each sub-basin for different flood events. The mean value and the standard deviation are provided in lower rows.
**Table 6.** Manning's coefficients of the numerical domain.

**Table 1.** Location and elevation of the rain gauges located in the area of study (The system of reference for latitude and longitude is the EPSG: 4326).

| Rain gauge id. | Name | Latitude | Longitude | Elevation (m.a.s.l.) |
|---|---|---|---|---|
| $rg_1$ | Labrada | 43.4054 | -7.50205 | 662 |
| $rg_2$ | Lanzós | 43.3746 | -7.64468 | 470 |
| $rg_3$ | Guitiriz-Mirador | 43.2266 | -7.78307 | 684 |
| $rg_4$ | Sanbreixo | 43.1457 | -7.79112 | 496 |
| $rg_5$ | Castro de Rei Lea | 43.1559 | -7.48588 | 428 |
| $rg_6$ | Pol | 43.1626 | -7.28258 | 647 |
| $rg_7$ | Corno do Boi | 43.0374 | -7.89265 | 731 |

**Table 2.** Values of the correlation (Spearman's r) and normalised standard deviation ($\sigma_n$) of the precipitation forecast using the measured data as reference at each rain gauge, considering the complete time series of precipitation. The averaged values for each precipitation forecast are also shown.

| | Forecast window (h) | | | | | |
|---|---|---|---|---|---|---|
| | 1-24 | | 25-48 | | 49-72 | |
| Rain gauge | r | $\sigma_n$ | r | $\sigma_n$ | r | $\sigma_n$ |
| $rg_1$ | 0.84 | 0.80 | 0.82 | 0.81 | 0.77 | 0.80 |
| $rg_2$ | 0.84 | 1.09 | 0.82 | 1.07 | 0.79 | 1.07 |
| $rg_3$ | 0.83 | 1.00 | 0.81 | 0.96 | 0.77 | 0.99 |
| $rg_4$ | 0.81 | 0.97 | 0.79 | 0.96 | 0.75 | 0.98 |
| $rg_5$ | 0.81 | 1.13 | 0.80 | 1.10 | 0.76 | 1.12 |
| $rg_6$ | 0.84 | 1.16 | 0.83 | 1.07 | 0.79 | 1.07 |
| $rg_7$ | 0.83 | 1.05 | 0.81 | 1.06 | 0.77 | 1.10 |
| Mean value | 0.83 | 1.03 | 0.81 | 1.00 | 0.77 | 1.02 |

**Table 3.** Values of the correlation (Spearman's r) and normalised standard deviation ($\sigma_n$) of the precipitation forecast using the measured data as reference at each rain gauge, considering only rainy events (above the 75th percentile). The averaged values for each precipitation forecast are also shown.

| | Forecast window (h) | | | | | |
|---|---|---|---|---|---|---|
| | 1-24 | | 25-48 | | 49-72 | |
| Rain gauge | r | $\sigma_n$ | r | $\sigma_n$ | r | $\sigma_n$ |
| $rg_1$ | 0.66 | 0.72 | 0.61 | 0.70 | 0.53 | 0.72 |
| $rg_2$ | 0.71 | 1.00 | 0.63 | 0.98 | 0.56 | 0.99 |
| $rg_3$ | 0.70 | 0.98 | 0.61 | 0.93 | 0.59 | 0.98 |
| $rg_4$ | 0.73 | 0.93 | 0.65 | 0.90 | 0.60 | 0.93 |
| $rg_5$ | 0.68 | 1.02 | 0.63 | 1.01 | 0.54 | 1.04 |
| $rg_6$ | 0.69 | 1.14 | 0.65 | 0.98 | 0.56 | 1.00 |
| $rg_7$ | 0.74 | 1.03 | 0.68 | 1.02 | 0.63 | 1.10 |
| Mean value | 0.70 | 0.97 | 0.64 | 0.93 | 0.57 | 0.97 |

**Table 4.** Main characteristics of the analysed flood events

| Date of the flood event | Duration (days) | Initial flow ($m^3s^{-1}$) | Initial depth (m) |
|---|---|---|---|
| 28/12/09 | 4 | 52 | 1.3 |
| 17/11/10 | 5 | 116 | 1.7 |
| 17/01/13 | 10 | 164 | 1.9 |
| 11/03/13 | 5 | 179 | 2.0 |
| 05/11/13 | 7 | 234 | 2.3 |
| 14/01/13 | 10 | 165 | 1.9 |
| 28/01/14 | 15 | 202 | 2.1 |
| 01/03/14 | 4 | 134 | 1.8 |
| 30/01/15 | 3 | 184 | 2.0 |
| 01/03/15 | 3 | 134 | 1.8 |
| 10/02/16 | 7 | 216 | 2.1 |
| 26/02/16 | 3 | 137 | 1.8 |
| 05/03/16 | 4 | 175 | 2.0 |
| 10/03/18 | 6 | 154 | 1.9 |
| 30/03/18 | 4 | 201 | 2.1 |

**Table 5.** Curve number ($CN$) and lag time ($L_g$) values for each sub-basin for different flood events. The mean value and the standard deviation are provided in lower rows.

| Date of the flood event | Sb$_1$ | | Sb$_2$ | | Sb$_3$ | |
|---|---|---|---|---|---|---|
| | $CN$ | $L_g$ (min) | $CN$ | $L_g$ (min) | $CN$ | $L_g$ (min) |
| 12/09 | 92 | 1154 | 97 | 2700 | 98 | 2770 |
| 11/10 | 80 | 1140 | 84 | 2702 | 80 | 2781 |
| 03/13 | 79 | 1157 | 96 | 2701 | 99 | 2774 |
| 11/13 | 80 | 1148 | 86 | 2685 | 83 | 2778 |
| 01/14 | 78 | 1155 | 96 | 2700 | 98 | 2767 |
| 03/14 | 81 | 1153 | 88 | 2706 | 92 | 2764 |
| 01/15 | 96 | 1153 | 99 | 2701 | 99 | 2773 |
| 03/15 | 81 | 1151 | 91 | 2700 | 98 | 2771 |
| 02/16 | 81 | 1155 | 88 | 2700 | 98 | 2767 |
| 03/16 | 82 | 1153 | 80 | 2711 | 84 | 2764 |
| 03/18 | 80 | 1152 | 82 | 2691 | 93 | 2769 |
| Mean | 85 | 1152 | 90 | 2700 | 93 | 2771 |
| σ | 6 | 4 | 6 | 7 | 7 | 5 |

**Table 6.** Manning's coefficients of the numerical domain.

| Land's uses | Manning's coefficient ($s\ m^{-1/3}$) |
|---|---|
| River | 0.025 |
| Brush | 0.050 |
| Trees | 0.120 |
| Sparse vegetation | 0.080 |
| Infrastructure | 0.020 |
| Industrial | 0.100 |
| Residential | 0.150 |

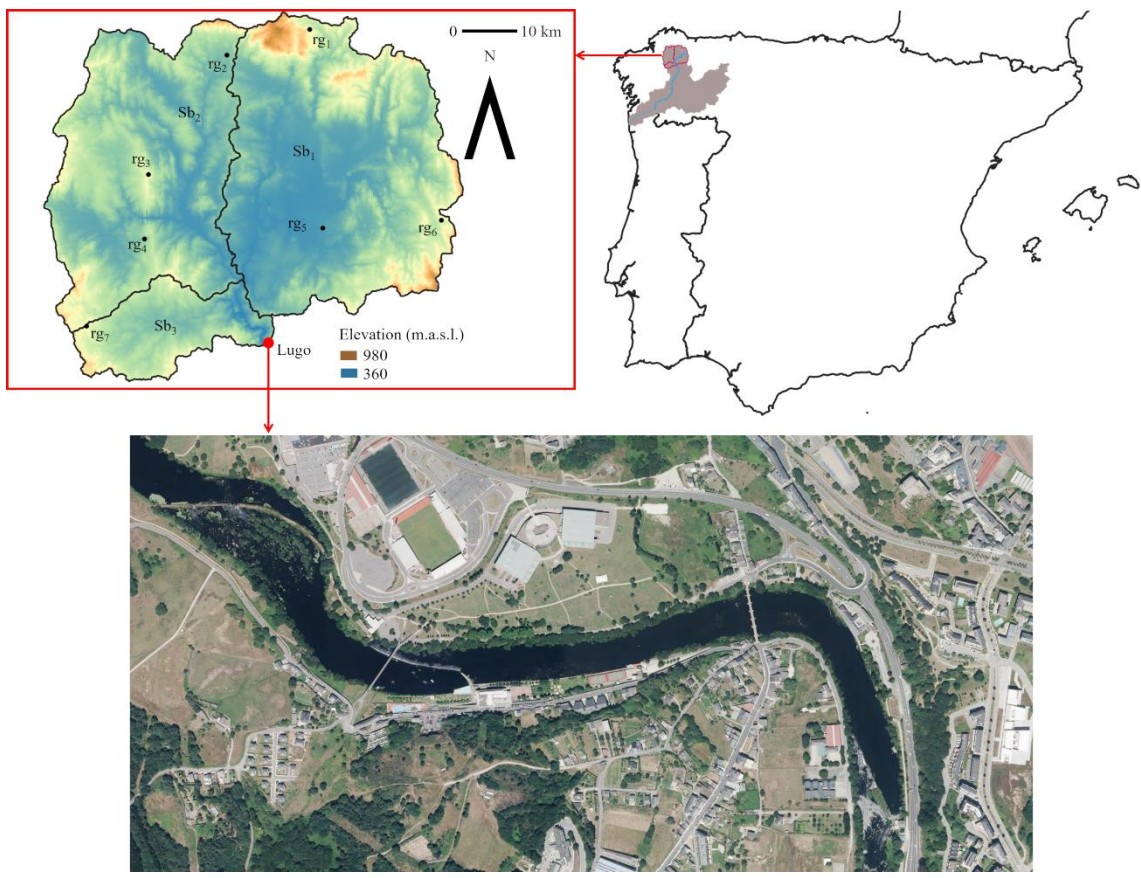


**Figure 1.** Area of study. In the upper right panel, the location of the entire catchment of
the shared Portuguese-Spanish river (shaded area) in the Iberian Peninsula and the
riverbed of the Miño river (blue line) are shown. The rain gauges (rg1, …,rg7) located in
the catchment and the sub-basins ($Sb_1$, $Sb_2$ and $Sb_3$) of the domain (upper left panel), as
well as the area of study in Lugo (lower panel) are also shown. (PNOA courtesy of ©
Instituto Geográfico Nacional).





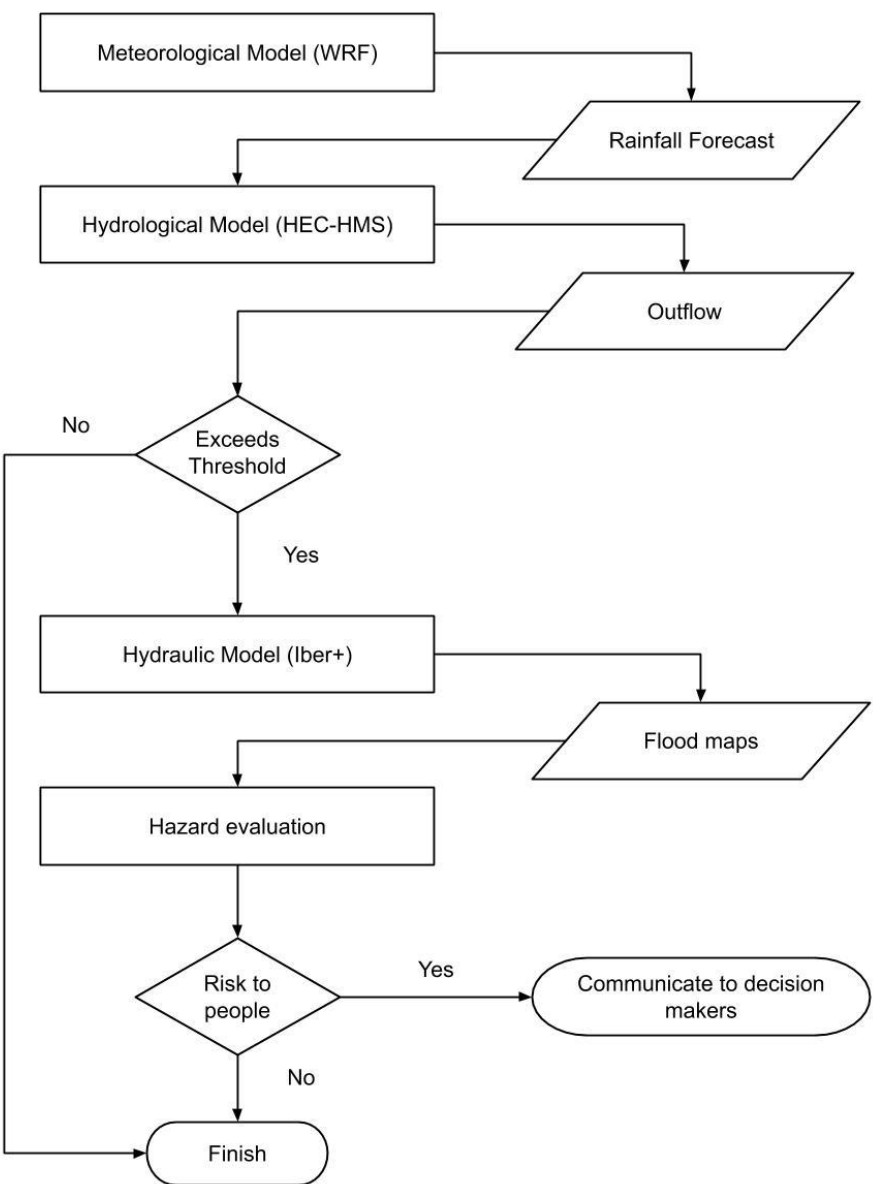


**Figure 2.** Flowchart of the proposed EWS.





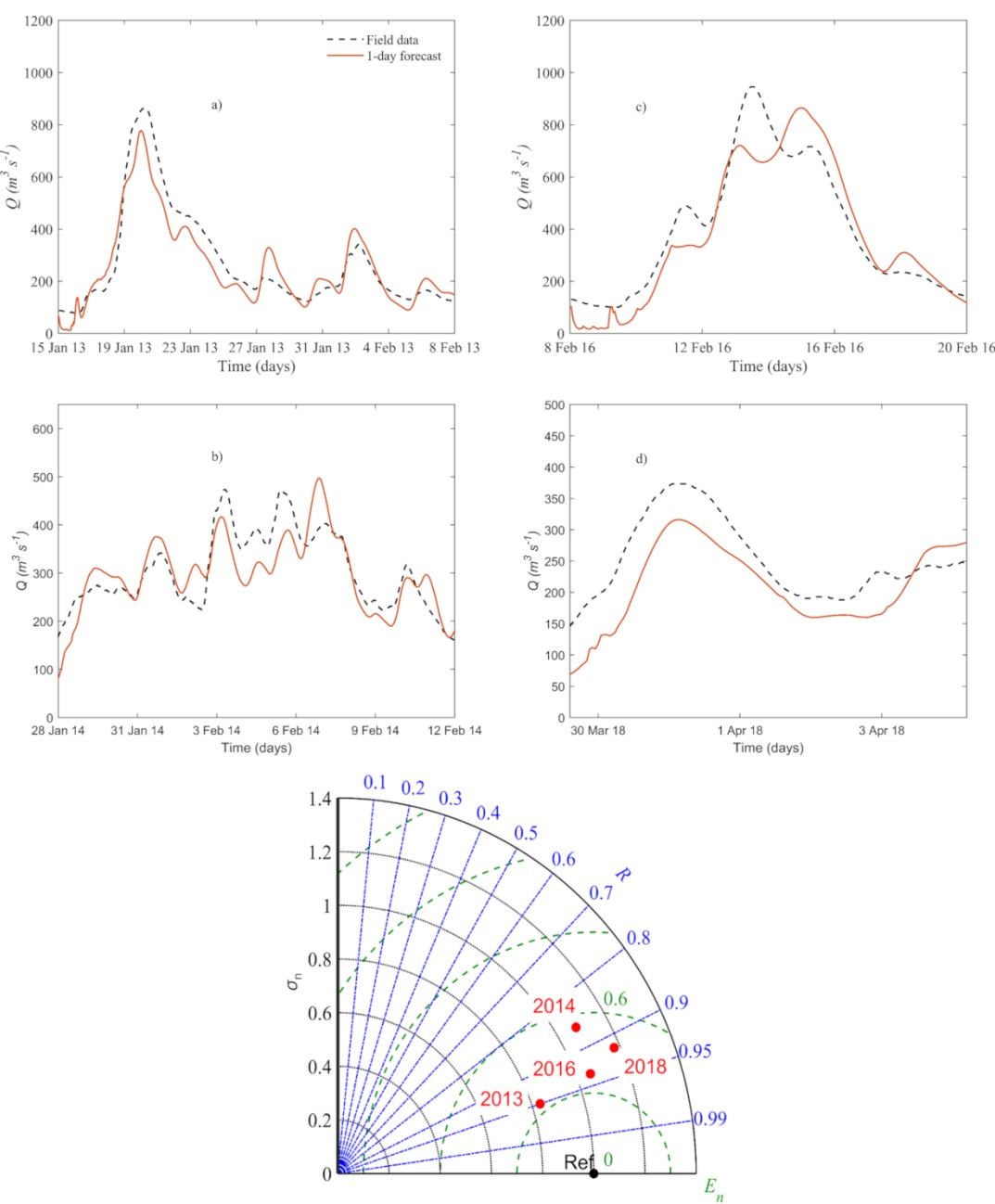


**Figure 3.** Time series of the registered streamflow (dashed line) and numerical
streamflow (orange line) of the validation events: a) 01/2013, b) 01/2014, c) 02/2016 and
d) 03/2018). Taylor diagram of the validation cases is also shown.

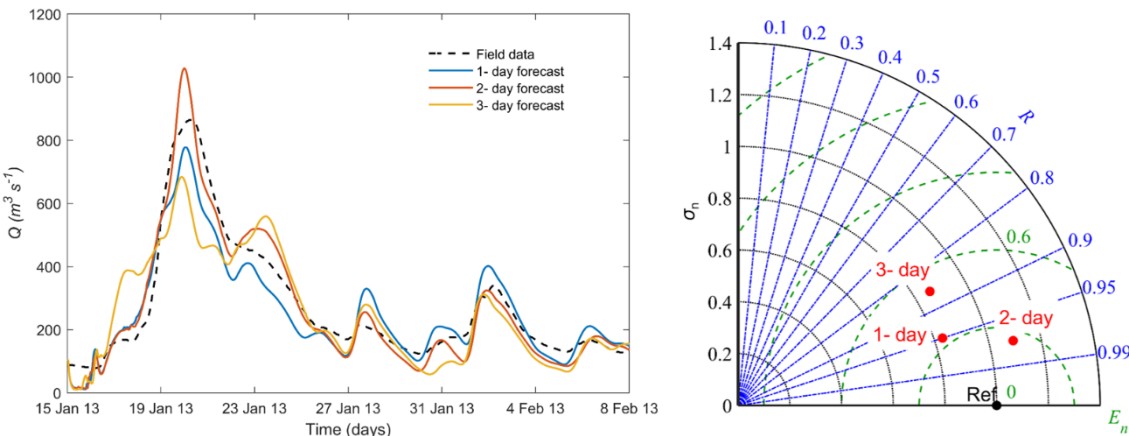


**Figure 4.** Time series of the outflow at the control point obtained in the gauge station
(dashed line) and calculated using the three forecast windows (left panel) and Taylor
diagram for the same cases (right panel).

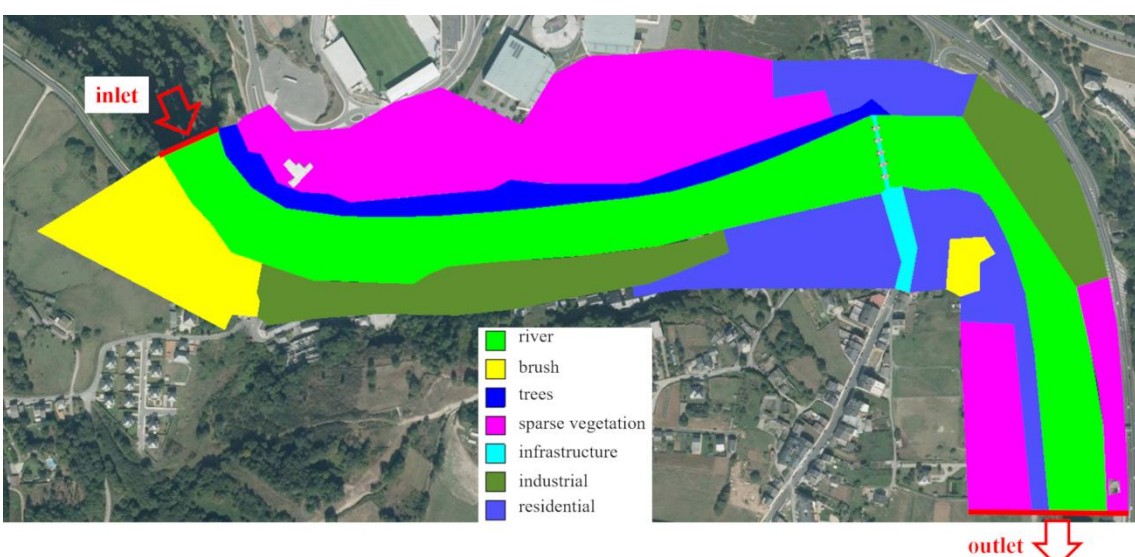


**Figure 5.** Numerical domain at Lugo. The land uses and the location of the boundary
conditions (red lines) are also shown. (PNOA courtesy of © Instituto Geográfico
Nacional).

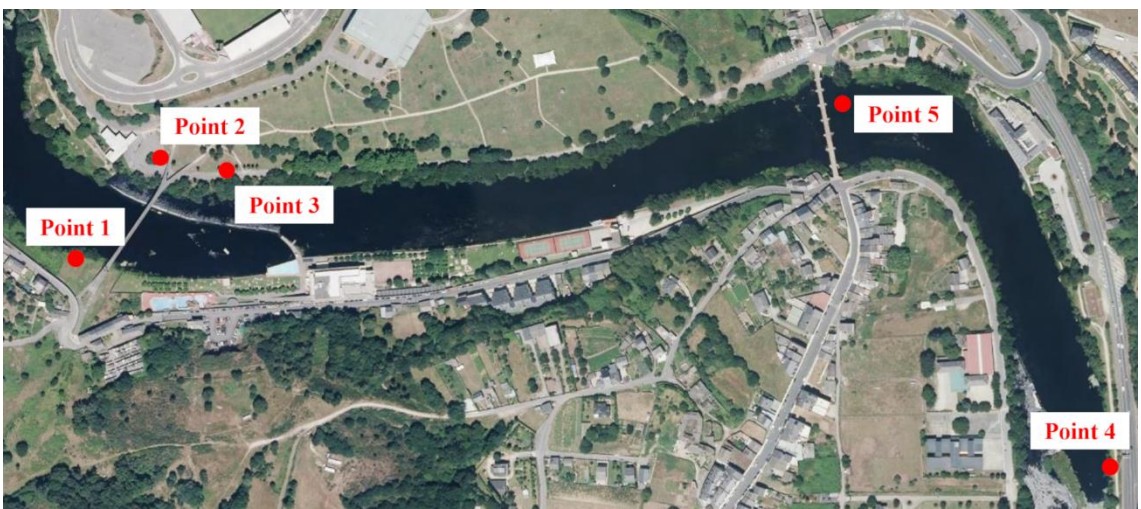

**Figure 6.** Location of the five control points at the area of study in Lugo. (PNOA courtesy of © Instituto Geográfico Nacional).

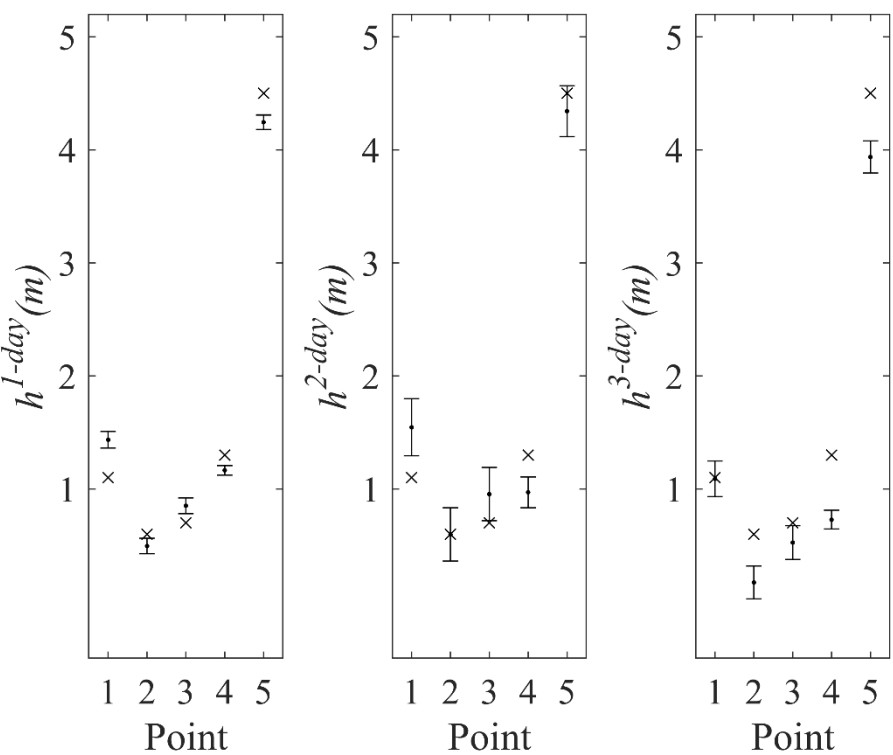

**Figure 7.** Comparison between water depth ($h$ in meters) between the numerical model (.) and the field data (x) for the three forecast windows 1-day (left), 2-day (middle) and 3-day (right). The range of the numerical values correspond to 3 times the standard deviation of the elevations obtained from 12:00 to 16:00 on January 20[th], 2013.


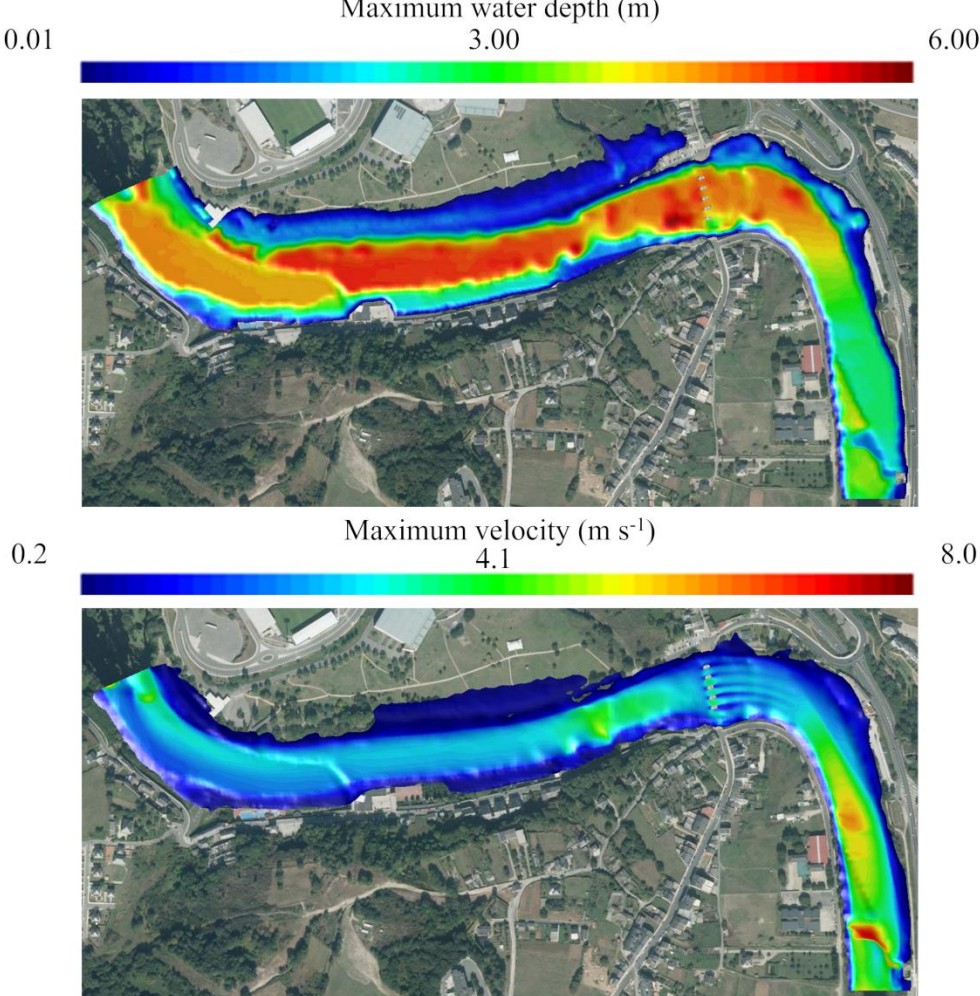



**Figure 8.** Maximum water depth (upper panel) and maximum velocity (lower panel) obtained with Iber+ for the 1-day precipitation forecast. (PNOA courtesy of © Instituto Geográfico Nacional).





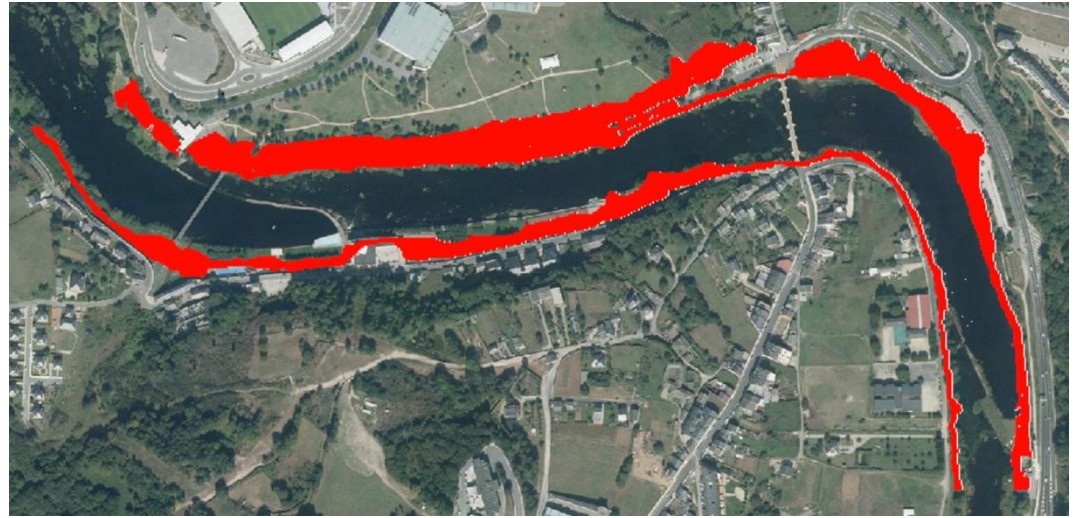



**Figure 9.** Areas where hazard criterion is surpassed. (PNOA courtesy of © Instituto Geográfico Nacional).
