# Peer review of "Towards an Automatic Early Warning System of Flood 1 Hazards based on Precipitation Forecast: The case of the 2 Miño River (NW Spain) 3 4 5 J. González-Cao, O. García-Feal, D. Fernández-Nóvoa, J.M. Domínguez-Alonso, M. Gómez-6 Gesteira 7 8 Environmental Physics Laboratory (EPhysLab), CI"

_Natural Hazards and Earth System Sciences, 2019_

## Referee Comment (RC1) · Anonymous Referee #1 · 26 Jul 2019

General comments

The authors present an Early Warning System (EWS) for Flood Hazards and its application to a watershed in Spain. This is an interesting topic within the scope of NHESS. EWS have been identified as an efficient approach for increasing resilience against flood damage reductions.

The paper is well structured and clearly written. The methodology is in general well described although some aspects (mentioned in the Specific comments) need some clarification.

As mentioned by the authors in the introduction, EWS have been implemented in re-

cent years at different scales and using different approaches. The authors should mention the differential aspects of the proposed Early Warning System (EWS) with respect to other systems that have been proposed, some of which are mentioned in the introduction (lines 45-75).

More details about the model implementation at the Miño basin should be given in lines 169-176. Some details that might be of interest to the reader are: infiltration model and parameters, formulation and parameters used to compute rainfall-runoff transformation, formulation and parameters used for runoff propagation in the stream network, number and size of subbasins, estimated characteristic times of the watershed and subbasins(as lag time or concentration time), is it baseflow considered? With which formulation?

Specific comments

HEC-HMS is a semi-distributed hydrological model, not a distributed model as stated in line 166. With the implementation of the authors (the whole watershed of 2200 km2 is divided in only 3 sub-basins) it should even be considered as an aggregated model.

How is the hydraulic inundation model initialised at the beginning of the flood hydrograph (with which discharge and/or water depths)?

In section 4.1. the authors discuss the accuracy of the precipitation forecasts. This is very interesting since it is one of the main reasons that might explain the performance of the whole system. The authors evaluate the accuracy for a rainfall aggregation time of 24 hours. I guess they have chosen this aggregation time taking into consideration the concentration time of the watershed. However, it would be interesting to show the accuracy of the precipitation forecasts for smaller aggregation times, as for instance 12 h, 6 h, 1 h, since those times are more relevant for smaller basins.

How does the calibrated Lag Time in Table 3 relate to empirical formulas based on the basin size and slopes?

Figure 7 shows a comparison between the water depth measured and computed at certain locations that are shown in Figure 6. Were those field values measured (and how)? Or are they values estimated by visual observation and/or photographs of the inundation? At what time during the flood were they measured/observed? Are they maximum values or values at a specific time? Those questions should be clarified in the paper.

In Figure 7 the authors present a range of variation of the numerical values, but this ranges are computed rather arbitrarily (3 times the standard deviation of the water depths from 12 am to 4 pm). Why has this criterion been chosen and how does it relate to model output uncertainty? If this is not discussed, the range bars shown in Figure 7 are meaningless for model evaluation since they can be as large or small as one wants by just changing the criterion.

How is the error in Table 5 defined and which is the probability of the real depth being inside the intervals given?

---

## Referee Comment (RC2) · Rodrigo Maia (Referee) · 27 Jul 2019

**General comments**

The paper presents the stage of the development of an early warning system based on precipitation forecast using a common hydrologic model (HEC-HMS) and specific hydraulic 2D model (Iber+), applied to the upper catchment of a shared Portuguese-Spanish river. The overall results and the quality of the paper are quite good and the future and wider application of the modelling System to flood risk mapping in urban areas, namely in Spain, looks bright.
A revision of English and namely of the verb tenses is suggested. Also choose following either British English or American English in the text (e.g., analysed (line 60) vs organized (line 90)).

**Specific comments**

Introduction:

1) A literature review on available and referenced flood early warning systems (short-term forecast) to frame and compare with the one proposed in the study is missing and is suggested.

2) Several references are made to flash flood studies (lines 49 to 53) and to flash flood EWSs (e.g., line 57). A clarification on the type and range of floods under study is required and suggested.

Study Area:

1) Figure 1 (right) locates it as an intermediate part of a non-identified/delimited supposedly full river catchment. This last shall be clearly identified as the Spanish part of the shared Miño river catchment, and the location (red square) shall be adjusted. Also, it is suggested that the location is done in a full map of the Iberian Peninsula.

2) Information on the type of flow regime is missing and is suggested.

Methodology:

1) The effect of the initial flow condition (namely for torrential regimes) in the calculations is not discussed in the text. Is that negligible? What was considered as the initial condition (e.g., the water level in the river) in the developed EWS (particularly, in the hydraulic model Iber+)?

2) Lines 127-128: "the criteria of Cox et al. (2010) are used to define safety limits …"
Why were those criteria considered suitable for that in this study? Accordance to EU regulations?

Results and discussion:

1) Concerning the accuracy of MeteoGalicia precipitation, data was evaluated by computing the Spearman correlation coefficient presented in Table 2. Nevertheless, the correlation is not enough to guarantee the accuracy and so some statistical indicator(s) would also be required.

2) Line 225-227: "Therefore, it can be concluded that the precipitation forecast provided by MeteoGalicia offers results very close to the real rain events for the entire time series of precipitation data (2008-2018)".

In fact, if hourly precipitation was used, the amount of null or very small values should be a very high percentage of the sample values. In order to predict floods, the test would require higher precipitation values (e.g. values above the 90th percentile) to be considered. To clarify/correct, see Brown et al. (2010).

Brown, J., Demargne, J., Seo, D-J., Liu, Y. (2010). The Ensemble Verification System (EVS): a software tool for verifying ensemble forecasts of hydrometeorological and hydrologic variables at discrete locations. Environmental Modelling and Software, 25(7), pp 854-872.

3) Concerning calibration and validation of HEC-HMS:

3.1) Line 230-231: "A set of 15 extreme flood events registered during the period 2008-2018 were used to calibrate and validate the rain-runoff model HEC-HMS.."
It is suggested that the time duration of all the events as also the initial flow conditions are referred and/or resumed.

3.2) Line 232-233: "…with HEC-HMS using the 1-day forecast of precipitation".

Q: Why calibrate for 1 day forecast and not use the real precipitation data? Wasn´t the accuracy of forecasted data checked before? If so, shouldn't also be done a calibration for 2 and 3 days forecast windows?  To clarify/justify

3.3) Lines 240- 241: "The mean values of CN and Lg of each sub-basin were used to validate the model in four flood events (01/2013, 01/2014, 02/2016 and 03/2018) by means of a Taylor diagram."
So, that meaning that those values are to be used for any simulations to be performed. So, the initial flow conditions and/or flood time duration are not relevant?  That should be addressed and/or justified.
The presentation of the results for all four flood events (and not only for one) could contribute to that and is suggested.

3.4) Lines 245-247: "These (statistical indicators) show that the mean values of CN and Lg obtained in the calibration step characterise the behaviour of the basin with a high accuracy"
That could possibly be better justified, namely by means of referenced bibliography.

4) Case Study:
4.1) Authors are recommended to present more details about the hydraulic model setup.
   - What type of boundary conditions were assumed at the inlet and outlet sections?
   - How were the turbulence flow features modeled in this study?
   - Did authors perform any sensitivity analysis on the grid system or the Manning's coefficients used?
   - Were (all) the bridges considered in the study? If so, how? (Figure 8, bottom, shows bridge pier effects at one bridge).

4.2) Line 264: What is the resolution of the topography data used in this study?

**Technical corrections**

Line 20: models

Line 29: estimated?

Line 49: enhanced?

Line 52: analysed? Lines 55-58: The sentence requires revision.

Line 79: to predict extreme flows?

Lines 87-85: "This model was *previously* calibrated for the area of study by means of series of historical flood events detected over the last decade."

"This model was calibrated for the area of study by means of series of historical flood events detected over the last decade." ?

Line 85: Iber+?

Figure 1: Symbols Sb1, .. Sb3 were not defined in the text (sub-basins?).

Sugg: Add the scale and north direction, particularly in the sub-figure including the city of Lugo: include Portugal (and the sea).

Line 101: This sub-catchment area?

Line 106: in the entire sub-catchment?

Line 111: also affects?

Line 131: Magnitude velocity

Line 135: "The details of the components of the EWS …. are  described in the following sections." presented?

Line 141: The link requires an update.

Line 154: Data from these rain gauges was used to assess the …?

Line 155: "The rain gauges selected for this study were shown in Figure 1.."

Sugg: The mentioned rain gauges are pictured … .

Line 158: "Minho River"

Miño river

Line 167: Iber+?

Line 178: …normalised centred …

Line 181: meaning of subscript $n$ is missing

Line 181-185: equations (1) to (3): i = 1?

Line 190: Iber+?

Line 191: 2D (Two-Dimensional) (first time use of the abbreviation in the text)

Line 200: GPU: Graphics Processing Unit?

Line 205: (Definition for *bias* parameter?).

Line 244: "root mean square difference" (RMSD?) (see line 252)

Q: RMSD or RMSE (line 205)

Figure 8: Color-map is not clearly informative (e.g.: what are the values corresponding to the green color range?)

---

## Author Comment (AC1) · 26 Sep 2019

Authors attach a zip file including the response to the reviewer's comments ("Response_to_Reviewers.pdf") , as well as the new version of the revised manuscript ("Gonzalez_Cao_et_al_Revision.pdf") in order to clarify some explanations included in the response to reviewers.

Please also note the supplement to this comment:
https://www.nat-hazards-earth-syst-sci-discuss.net/nhess-2019-200/nhess-2019-200-AC1-supplement.zip

---

## Author Response (AR1)

Editor Comments:

I am pleased to inform you that your paper has been accepted for publication. The revision is a substantial improvement on the original. Please check this technical corrections:

Page 1 line 32 please update the web address https://www.nws.noaa.gov/om/hazstats.shtml to https://www.weather.gov/hazstat/ (check all the web address).

**Done.**

Page 2 line 43 replace [UN, 2006] by (UN, 2006).

**Done.**